# Seismic data acquisition for combining high-resolution seismic reflection and full-waveform inversion – a case study for overdeepened valleys

Thomas Burschil[1], Daniel Köhn[2], Matthias Körbe[3], Gerald Gabriel[3,4], Johannes Großmann[5], Gustav Firla[6], Markus Fiebig[6]

[1]Federal Institute for Geosciences and Natural Resources (BGR), Hannover, 30655, Germany
[2]Kiel University, Kiel, 24118, Germany
[3]LIAG Institute for Applied Geophysics, Hannover, 30655, Germany
[4]Institute of Earth System Sciences, Section Geology, Leibniz University, Hannover, 30655, Germany
[5]Bavarian Environment Agency (LfU), Hof, 95030, Germany
[6]Department of Landscape, Water and Infrastructure, BOKU University, Vienna, 1190, Austria

*Correspondence to*: Thomas Burschil (thomas.burschil@bgr.de)

**Abstract.** In the context of the ICDP-project "Drilling Overdeepened Alpine Valleys", the integration of high-resolution seismic reflection (HRSR) and full waveform inversion (FWI) aims to enhance detailed near-surface imaging in heterogeneous glacial and post-glacial environments. To meet the specific requirements of both methods, dense P-wave and S-wave datasets were acquired over an overdeepened basin in the northern Alpine foreland, using a combination of vibratory and explosive seismic sources as well as different receivers. Analysis of the datasets as well as a separate application of HRSR and FWI demonstrates the suitability of the acquired data. The FWI workflow applied to the P-wave data fits the recorded waveforms and converged successfully, yielding a consistent and physically reasonable model that correlates well with the HRSR images. These datasets are the basis for future methodological development of combining HRSR and FWI.

Furthermore, HRSR images reveal important information about the geology of the overdeepened basin near the town of Schäftlarn (Germany). Using standard processing workflows, P-wave HRSR delineates detailed subsurface structures, including the base of the basin, intra-basin discontinuities, and subtle stratigraphic variations. Additional S-wave data provides superior resolution in imaging Quaternary basin sediments down to 200 m depth compared to P-waves and thus offers complementary information.

## 1 Introduction

Near-surface seismic methods are highly effective in subsurface imaging, for example in detecting stratigraphic features and geological discontinuities (Wang et al., 2025). In heterogeneous near-surface environments such as glacial and post-glacial deposits, seismic imaging can be challenging (Maraio et

al., 2018). Nonetheless, even in such environments high-resolution seismic reflection method (HRSR)
provides detailed structural information by analyzing the reflected wavefield (e.g., Dehnert et al., 2012;
Malehmir et al., 2013; Maries et al., 2017). Depending on the survey parameters, this method is often
suitable to image the subsurface structure from depths of a few metres with a resolution in the sub-metre
range down to several kilometres with decreasing resolution. Utilizing S-waves instead of convenient P-
waves can increase resolution even more (Pugin et al., 2009; Burschil and Buness, 2020; Pertuz and
Mahlemir, 2023). However, the complexity of the subsurface is decisive for the required extent of
seismic data processing to ensure best imaging quality. Depending on the geological situation, true
amplitude processing using common reflection surfaces as well as prestack depth migration can be
superior to less advanced processing workflows and, thus, reveal undiscovered geological features that
are not observable after standard processing. Examples in the context of the investigation of
overdeepened structures are allochthonous Molasse blocks at the base of the sedimentary succession
(Burschil et al., 2018) or cuspate-lobate folding of shallow diamict (Buness et al., 2022).

Full-waveform inversion (FWI) further fertilizes seismic imaging (Tarantola, 1986; Virieux and Operto,
2009), enabling a detailed and quantitative reconstruction of subsurface properties (e.g., Operto et al.,
2013; Mecking et al., 2021; Singh et al., 2022; Beraus et al., 2024). With elastic FWI, it is possible to
gain information about S-wave velocities from P-wave data as well (Pan et al., 2019; Roodaki et al.,
2024). However, both methods have limitations: HRSR relies on impedance contrasts and shallow
reflections may be superimposed by near-surface scattering (Frei et al., 2015; Sloan et al., 2016), while
FWI is computationally intensive (Ren and Liu, 2015) and highly sensitive to initial model accuracy
(Zhang et al., 2025). The applicability of the latter depends heavily on high-quality field data and initial
models (Vigh et al., 2018). For elastic FWI, this includes initial P-wave and S-wave models. Despite
these challenges, we expect that their combined application significantly improves geophysical
interpretations in near-surface investigations, leading to better resolution, accuracy, and reliability in
data analysis. HRSR and FWI set different requirements on the raw data. HRSR needs a broad
bandwidth of the source signal to gain sharp reflections (Brodic et al., 2021). State-of-the-art FWI
workflows invert the data iteratively in stages, starting with a low frequency band to avoid cycle
skipping (Dokter et al., 2017; Vigh et al., 2018; Köhn et al., 2019). These complementary prerequisites
cause challenges for the acquisition of field data that is to be analyzed by an integrated approach.

To overcome these challenges, different land seismic sources are available. Each of them has certain
advantages and disadvantages in emitting seismic waves. Vibratory sources are often used since the
signal is highly repeatable and the vibrators emit the energy over a certain time length. The total emitted
energy accumulates but the instantaneous emitted energy is low so that the impact on the ground is low
and damage to land or infrastructure is avoided. The repeatability of emitting the same signal with the
same frequency content is high (Brodic et al., 2021) and advantageous for vertical stacking to reduce
incoherent noise. The peak force of a seismic vibrator refers to the maximum force output that the
vibrator can exert on the ground during operation (Sallas, 1984). The emitted sweep signal with a
defined frequency band is correlated with the recorded traces and produces a zero-phase Klauder-
wavelet by correlation with itself (Lines and Clayton, 1977). Depending on the frequency band and the
ground coupling, the emitted signal is close to the ideal, sharp Klauder-wavelet. Both advantages are

complemented by economic reasons, fewer regulations, and ease of approval. However, seismic vibrators are technically-limited in exciting low frequencies (Wei and Phillips, 2011). In contrast, impulsive sources emit the seismic energy instantaneously. These sources often contain a low frequency content, but depending on the emitted energy of the source, the depth of seismic imaging is limited.

Commonly used impulsive sources are sledgehammers, drop weights, and explosives with various charges. While sledgehammers and drop weights result in limited penetration depth, explosives deployed in boreholes are used for shallow and deep penetration (Denny and Johnson, 1991). However, using explosive source is not possible everywhere, for example in urban areas.

On the receiver side, geophones have been used since the early 20th century to record the ground motion (Dragoset, 2005). Depending on the damping, the resonance frequency gives a decreasing sensitivity for lower frequencies (Krohn, 1984). Cabled systems enable a direct control of the ground motion and, thus, quality control. Autonomous systems become increasingly affordable and stable so that the failure or loss of a few percent of receivers still enables a high fold (e.g., Manning et al., 2019;

Ourabah and Chatenay, 2022).

To gain adequate datasets for the methodological development of combining HRSR and FWI, we acquired field data with different source-receiver combinations. We use explosive sources to excite low frequencies and vertical vibrator sources for a signal with a broad bandwidth. Densely spaced vertical

geophones recorded the ground motion. Additional autonomous 3-component geophones with a lower resonance frequency and sparse spacing receive the low frequency content of the ground motion. Horizontally oriented vibrator and receivers complement the data to provide initial S-wave velocity models for FWI.

The first objective of this paper is to present the field datasets and show that these meet the prerequisites of HRSR and FWI. Results of each method, processed separately (HRSR using P-wave and S-wave as well as FWI), provide insights into the acquired datasets and demonstrate that they are suitable for future methodological development of combining HRSR and FWI for overdeepened valleys. The second objective of this paper is to image the structure of an overdeepened basin, close to the town of

Schäftlarn (Germany). The study area is located in the Alpine foreland, about 30 km south of Munich. It is part of a complex system of overdeepened valleys and basins that are widely spread across the European Alps (Preusser et al., 2010) and one study site of the ICDP project Drilling Overdeepened Alpine Valleys (DOVE).

## 2 The DOVE project and study site Schäftlarn

The ICDP project Drilling Overdeepened Alpine Valleys (Anselmetti et al., 2022) investigates glacially overdeepened structures on a pan-Alpine basis using drill cores and geophysical surveys. The pinpoint information gained by the cores is extrapolated by the geophysical surveys to 2-D and 3-D. Previous investigations focusing on overdeepened structures were limited to a local/regional scope. DOVE aims to gather a comprehensive picture of overdeepened structures on the scale of a whole mountain range.

The core research questions to be investigated revolve around the timing and extent of Middle
Pleistocene glaciations and the sedimentary dynamics associated with them.

Glacial erosion sculpted not only the high Alpine regions but also the foreland. The study site is located
in the northern Alpine foreland that was influenced by repeated Pleistocene glaciations (Preusser et al.,
2010). About 30 km south of Munich (Germany), the former Isar-Loisach glacier lobe excavated an
overdeepened basin, which Jerz (1979) described as a branch basin of the Wolfratshausen Basin to the
south. The overdeepened basin is located at the morphologically defined ice-marginal position of the
Last Glacial Maximum (LGM; Fig. 1a). The local bedrock consists of Upper-Freshwater Molasse
sediments and the basin is filled with Quaternary sediments. To the west of the study site, Lake
Starnberg and Lake Ammersee provide examples of overdeepened basins not entirely filled by
sediments. The study site is located on the southern margin of the Munich gravel plain ("Münchner
Schotterebene"; Jerz, 1993). The western area of the study site is elevated approximately 100 m above
the recent incision of the Isar valley in the east (Fig. 1b). The Molasse bedrock has been identified in
outcrops at the base of the Isar valley slope (Jerz, 1987).

At the DOVE site Schäftlarn (ICDP site 5068_3), the Bavarian Environment Agency drilled a research
borehole (5068_3_A) in 2017 and conducted a seismic refraction survey in 2018. The 198.8 m long
drill-core (Fig. 1c) shows the sedimentary sequence from bottom to top: $(A_1/A_2)$ ~83 m of fine-grained
sediments, (B) ~111 m of coarse-grained sediments, and (C) ~4 m of diamictic sediments. Remnants of
a basal diamict were recovered, but the bedrock was not reached (Firla et al., submitted). The refraction
survey did not image beneath the coarse-grained sediments.

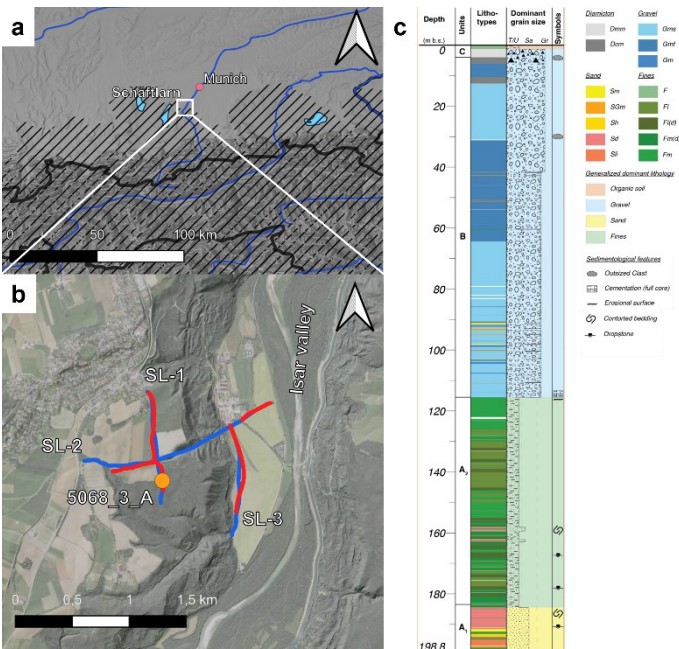


**Figure 1: Study area. (a) Overview map showing the location at the margin of the LGM ice extent (hashed area; Ehlers et al., 2011) and (b) location map with borehole ICDP 5068_3_A (orange) and seismic P-wave (blue) and S-wave (red) profiles. (c) Core description of borehole 5068_3_A after Firla et al. (submitted).**

# 3 Data acquisition

At the study site, we acquired data along three seismic profiles (SL-1 to SL-3) using P-waves and S-waves (Tab. 1; Fig. 1b). The layout was chosen to image the basin inventory as well as its base, which is estimated at ca. 200 m below the surface. For each seismic profile, we appended an additional suffix to indicate the wave type: 'P' for P-waves and 'S' for S-waves (for example SL-1P for the P-wave section and SL-1S for the S-wave section). Profile SL-2S is two-parted since the central part of SL-2

was not accessible for the S-wave survey.

**Table 1: Specifications of seismic profiles, including the number of vibrator points (VP), shot points (SP) of explosive sources, and receiver points (RP).**

| Profile | P-wave | | | | S-wave | | |
|---------|--------|--------|--------|--------|--------|--------|--------|
| | length | # of VP | # of SP | # of RP | length | # of VP | # of RP |
| SL-1 | 1020 m | 211 | 9 | 408 | 960 m | 250 | 408 |
| SL-2 | 1835 m | 331 | 11 | 468 | 580 m | 149 | 579 |
| | | | | | 300 m | 81 | 299 |
| SL-3 | 1020 m | 210 | 5 | 408 | 800 m | 208 | 408 |

## 3.1 Seismic sources

To excite P-waves, we used the 4-ton hydraulically driven vibrator MHV4P (Fig. 2a) of the LIAG Institute for Applied Geophysics, which has a vertical shaking unit (Burschil et al., 2021). This vibrator has a peak force of 30 kN and is limited to a lowest frequency of 20 Hz. In areas that were not accessible to vehicles (the central part of SL-2P), we used the electrodynamic vibrator ELVIS-7 (Wadas et al., 2016) with a vertical shaking unit (Fig. 2b). This wheelbarrow-mounted source, with a 1 kN peak

force, can also be deployed along difficult paths. For both sources, we set a dense vibrator point spacing (5 m), which was doubled the receiver spacing (2.5 m) along the profile. This spacing has proven to be efficient for shallow investigations in previous studies (e.g., Tanner et al., 2015; Wadas et al., 2016; Burschil et al., 2018). A 12-s sweep with linearly increasing frequencies of 20-200 Hz was used. In addition, we were able to conduct explosions using a total of 1 kg charges per shot location (Fig. 2c),

deployed in four 2-m deep boreholes per shot point (Fig. 2d). A hydraulic breaker on a mid-sized excavator was used to push a 2-meter-long metal rod into the ground to create the boreholes. We used a metal plate with a 1-by-1-m jig for positioning. For the entire acquisition, only 26 explosive source points were feasible due to logistic and financial constraints. As an S-wave source, we used the ELVIS-7 vibrator with a horizontally shaking unit positioned perpendicular to the profile direction. The

nominal vibrator point spacing was 4 m (four times the receiver spacing of 1 m), which was a

compromise regarding the fold and the measuring process. A 12-s linear sweep with frequencies of 20-120 Hz was used for the S-wave surveys.

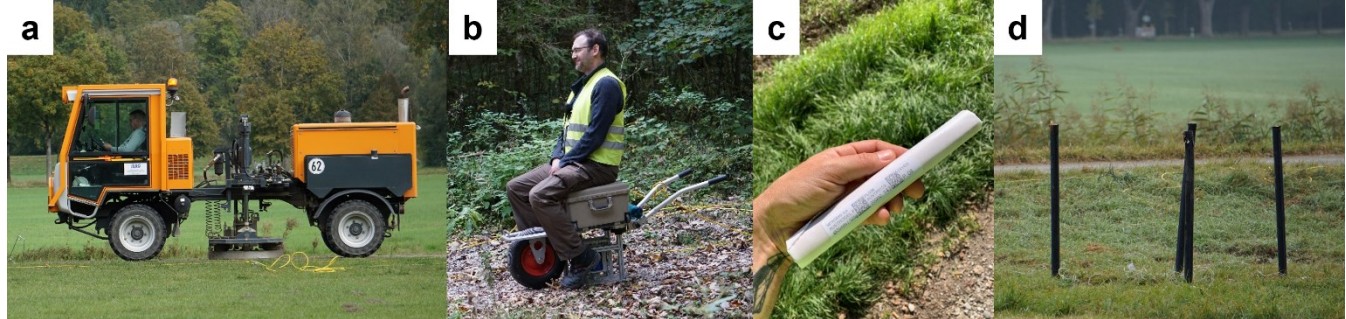

**Figure 2: Seismic sources. (a) Vibrators MHV4P, (b) ELVIS-7, (c) explosive charge, and (d) boreholes prepared for charging.**

## 3.2 Recording, receivers, and layout

For the P-wave surveys, 17 Geometric Geodes with 24 channels were used for recording (Tab. 2). As receivers, we deployed 408 vertical single-component 20-Hz geophones (Fig. 3a). The geophones are connected via cable to the Geodes. Each Geode correlated the signal using a pilot sweep and transferred
the digitized data to the recording vehicle via a wired network. The nominal receiver spacing was 2.5 m to ensure good coverage and to achieve a CMP bin size of <2.5 m to avoid spatial aliasing (cf. Buness et al., 2022). For the layout, we chose a single spread for SL-1P and SL-3P. For profile SL-2P, we used a split-spread layout with roll-along geometry. The roll-along distance was chosen such that the maximum offset was at least sufficient to image the expected basin depth of ~200 m (cf. Burschil, 2024
for more details).

In addition, for the P-wave survey, we deployed 28 autonomous Omnirecs DATA-CUBE[3] recording units along the profiles (Fig. 3b). Three-component 4.5-Hz geophones were connected to each DATA-CUBE³. These receivers were intended to detect lower frequencies than the 20-Hz geophones more effectively. These geophones were placed at approximately 40-60 m spacing as a single spread.

For the S-wave survey, we used 10 Geometric Geodes connected to two landstreamers for recording (Fig. 3c). Each landstreamer consists of 120 horizontal 10-Hz geophones with 1 m spacing, recording ground motion perpendicular to the profile direction. For the layout, we used a split-spread layout with roll-along geometry as in previous surveys (Burschil and Buness, 2020).

We refer to these datasets on the receiver side as geode-data, cube-data, and landstreamer-data,
respectively.

**Table 2: Specifications of seismic receivers.**

|  | Recording | Geophones | Orientation | # of channels | Nominal spacing | Mode |
|---|---|---|---|---|---|---|

| geode-data | Geometrics Geode | Sensor SM-6, 20 Hz | vertical | 408 | 2.5 m | triggered |
|---|---|---|---|---|---|---|
| cube-data | DATA-CUBE[3] | HL-6B, 4.5 Hz | 3-component | 3x 28 | ~40-60 m | autonomous |
| landstreamer-data | Geometrics Geode | Sensor SM-6, 10 Hz | horizontal | 240 | 1 m | triggered |

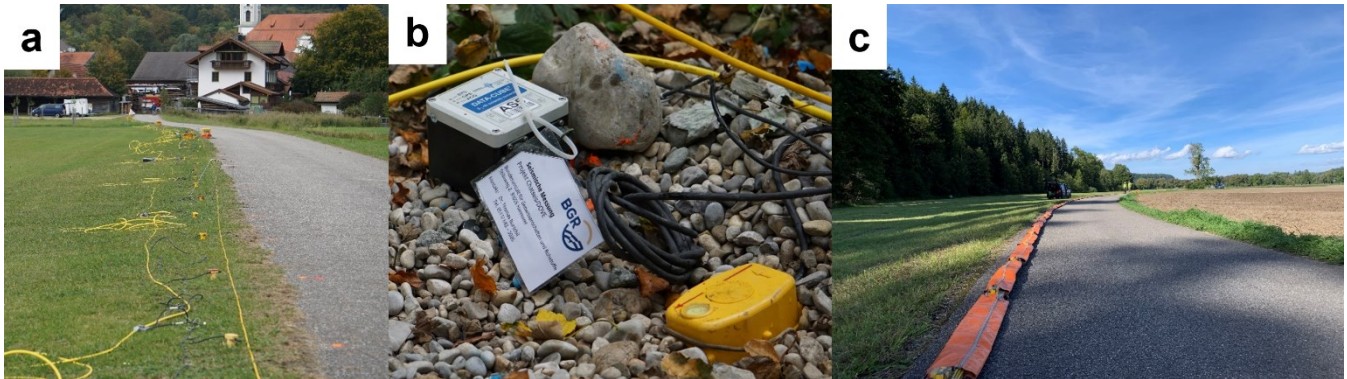

**Figure 3: Seismic receivers. (a) Planted vertical geophones connected by cable, (b) 3-component geophone and DATA-CUBE³, and (c) landstreamer with horizontal geophones.**

## 4 Data processing

HRSR processing and FWI were performed for different datasets so far (Tab. 3). The cube-data will contribute to combining HRSR and FWI in a future step.

**Table 3: Processing of different datasets. The cube-data are not processed yet.**

| | | Source | | |
|---|---|---|---|---|
| | | Explosives | Vertical vibrators | Horizontal vibrator |
| Receiver | geode-data | FWI | HRSR | |
| | cube-data | X | X | |
| | landstreamer-data | | | HRSR |

## 4.1 HRSR processing

P-wave seismic processing was carried out using Landmark ProMAX/SeisSpace Version 5000.11.0.0. For all profiles, we used the same processing workflow and parameters, following previous studies (e.g., Dehnert et al., 2012; Burschil et al., 2018). The workflow comprised several processing steps

(Tab. 4). Step P5a was only applied to profile SL-2P. The combination of elevation static correction and
residual statics worked sufficiently so that no refraction static correction was needed. Step P10 provides
a stacking velocity field for every 25th CMP that we converted and adapted for migration (step P17)
and time-to-depth conversion (step P20).

**Table 4: Seismic P-wave processing steps.**

| Processing step | | Parameter |
|---|---|---|
| P1 | Trace editing | |
| P2 | Vertical stacking for noise suppression | |
| P3 | Geometry assignment | CMP spacing 1.25 m |
| P4 | True amplitude recovery | spherical divergence correction with 1-D velocity distribution |
| P5 | Minimum phase transformation | with adapted wavelet |
| P5a | Match filtering between MHV4P and ELVIS-7 source | for SL-2P only |
| P6 | Surface consistent spike deconvolution | type: spike, operator length 140 ms |
| P7 | Adaptive deconvolution (L2 norm spiking) | type: spike, operator length 80 ms |
| P8 | F-K filtering for near offsets | -100 m – 100 m, to eliminate chevron patterns |
| P9 | Elevation static correction | datum 680 m, correction velocity 1400 m/s |
| P10 | Two iterations of velocity analysis in combination with residual statics | Every 25th CMP |
| P11 | Automatic gain control | 250 ms window length |
| P12 | Normal moveout correction | 40 % stretch mute |
| P13 | Common-midpoint stacking | shift to final datum |
| P14 | Bandpass filtering | Ormsby filter, 50-60-170-190 Hz |
| P15 | F-X deconvolution | 80 ms window length, 40-450 Hz |
| P16 | Automatic gain control | 500 ms window length |
| P17 | Poststack FD time migration | angle <45° |
| P18 | Bandpass filtering | Ormsby filter, 50-60-170-190 Hz |
| P19 | Automatic gain control | 500 ms window length |
| P20 | Time-to-depth conversion | smoothed velocity field |


S-wave processing of horizontal-vibrator and landstreamer-data was adapted from the processing flow
discussed in Burschil and Buness (2020) and carried out using Shearwater Reveal Version 6.2. The
same workflow was used for all profiles. The processing steps are summarized in Table 5. Step S8
provides the stacking velocities, which were analyzed at least every 50 m, and subsequently smoothed
and converted to interval velocities using the Dix equation (Sheriff and Geldart, 1995). For S-wave
processing, we waived migration to reduce migration artifacts.

**Table 5: Seismic S-wave processing steps.**

| Processing step | | Parameter |
|---|---|---|
| S1 | Trace editing | |
| S2 | Vertical stacking | |
| S3 | Geometry assignment | CMP spacing 0.5 m |
| S4 | Surface-consistent corrections | in the source and receiver domains |
| S5 | Surface-consistent deconvolution | for sources and receivers |
| S6 | Bandpass filtering | time-variant Butterworth filter<br><400 ms: 50-60-100-120 Hz<br>>400 ms: 30-40-100-120 Hz |
| S7 | Automatic gain control | 200 ms window length |
| S8 | Apply residual statics using brute stack | |
| S9 | Several iterations of velocity analysis at a floating datum | guided by reflectors in the stacked section |
| S10 | Normal moveout correction at floating datum | 200% stretch mute |
| S11 | Common midpoint stacking | shift to final datum |
| S12 | Automatic gain control | 200 ms window length |


## 4.2 Full waveform-inversion

For FWI, we used the latest version of DENISE Black Edition (Köhn et al., 2012; 2014). This multiparameter FWI optimizes the P-wave and S-wave velocities as well as the density simultaneously, using adjoint state gradients within the L-BFGS optimization method (Liu and Nocedal, 1989). As input
data for the first FWI, we used the geode-data from the explosive sources. The raw data were preprocessed by using a 30-Hz low-cut filter. As the initial P-wave model, we created a 1-D gradient model based on first arrival analysis. The initial S-wave velocity model $v_S$ is estimated from the P-wave velocities $v_P$ by the following equation:

$$v_S = \frac{v_P}{\sqrt{3}} .$$ (1)

The density initial model $\rho$ is estimated by the empirical relation (Ulugergerli and Uyanik, 2007).

$$\rho = 1000 * (0.1055 * \log(v_S) + 1.3871).$$ (2)

From analyzing the Rayleigh surface wave, compared to the P-wave first arrivals, we inferred a significant influence of damping, so that we chose a visco-elastic modelling approach. To mitigate the non-linearity of the inverse problem, a sequential frequency inversion approach is applied. The FWI
workflow consists of a sequential frequency approach, inverting field data up to 7 Hz, 10 Hz, 15 Hz, 20 Hz, 25 Hz, and 30 Hz, respectively. The source wavelet for each source gather is computed by a stabilized Wiener deconvolution. Smoothness constraints are imposed by an anisotropic, spatial 2-D

Gaussian filter, whose length is adapted to the local P-/S-wavelength, and which is applied to the adjoint state gradients, respectively. In the x-direction, the gradients are smoothed over 1x local wavelength, and in the y-direction over 0.5x the local wavelength. Parameter cross-talk is mitigated by using quasi-Newton L-BFGS optimization together with parameter scaling, where the density updates are systematically decreased by a factor of 0.5x compared to the seismic-velocity model updates. Finally, a global correlation norm is used as an objective function to mitigate source-receiver coupling effects.

## 5 Results

### 5.1 Field data and spectral composition

The acquired data show an excellent data quality on the various source and receiver settings (Fig. 4). Explosive sources have a good coupling and penetration depth. The dense receiver spacing of the geode-data provide a clear first P-wave arrival, essential for the generation of an initial P-wave model for FWI (red arrows in Fig. 4). The explosive data show strong surface waves that are aliased in the sparse spacing of the cube-data (blue arrows). The vibrator source MHV4P generated excellent reflections (yellow arrows) and less surface waves than the explosive source data. An air wave arrival is also visible in the data (green arrow).

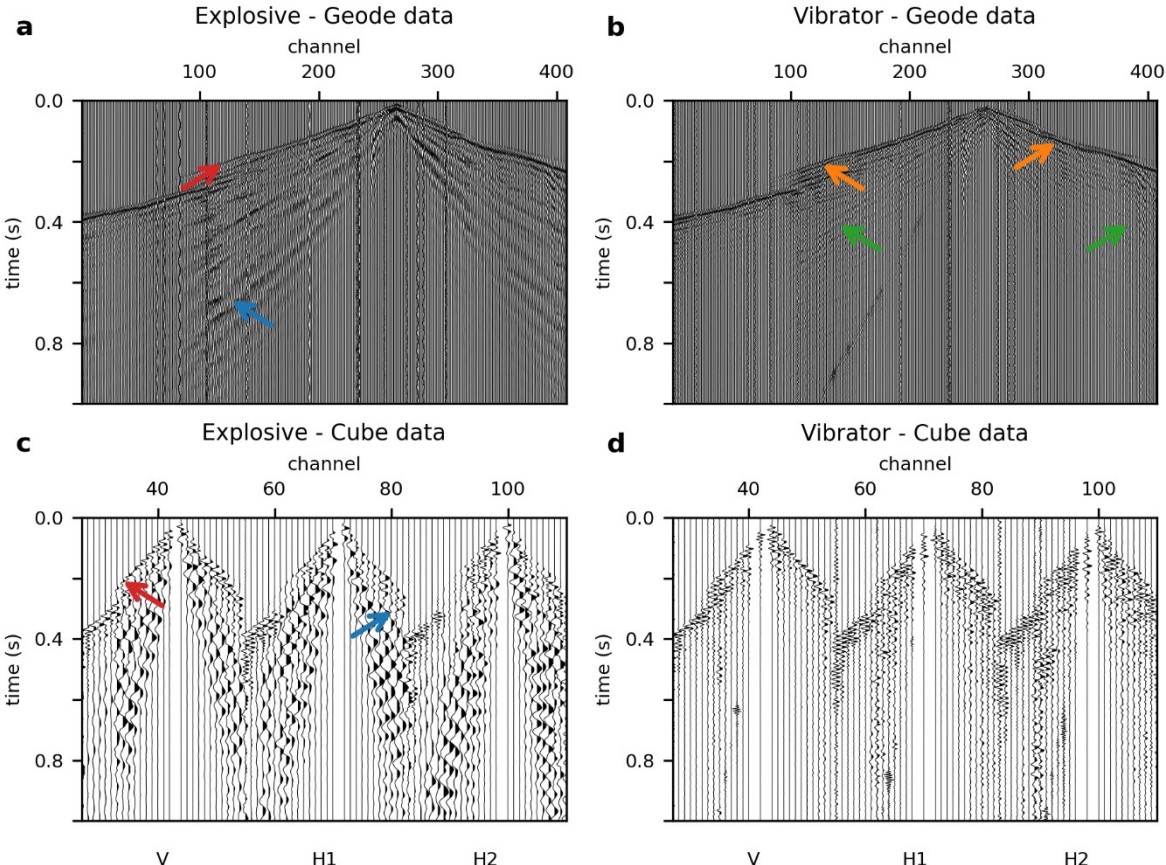

**Figure 4: Records for one source location (corresponding power spectral density in Fig. 5). (a) Explosive source and geode-data (20 Hz geophones), (b) vibratory source and geode-data (20 Hz geophones), (c) explosive source and cube-data (3-component 4.5 Hz geophones), (d) vibratory source and cube-data (3-component 4.5 Hz geophones). Note the strong surface waves (blue arrows) and the clear first arrivals (red arrows) in the explosive data, the high frequency reflections (orange arrows) in the vibratory data, as well as the air blast (green arrows).**

The frequency content of the P-wave data varies across the different source-receiver configurations (Fig. 5). The data of the vibrator source and geode-data (20 Hz geophones) show the entire sweep frequency range of 20-200 Hz. The corresponding cube-data (4.5 Hz geophones) show the same lower frequency ramp as the geode-data, but a high cut-off frequency around 160 Hz. The DATA-CUBE[3] have a maximum time sampling of 2.5 ms (400 Hz). The cut-off of higher frequencies is due to an anti-

alias filter in the device at 80% of the Nyquist frequency (160 Hz; T. Ryberg, priv. comm.). For the explosive sources, we detect a frequency content below 20 Hz in all data. Even the geode-data with 20 Hz geophones show a significant portion of energy below 20 Hz, the lower frequency flank looks similar to the cube-data with 4.5 Hz geophones. However, at these low frequencies the base level of the cube-data is much lower than for the geode-data.

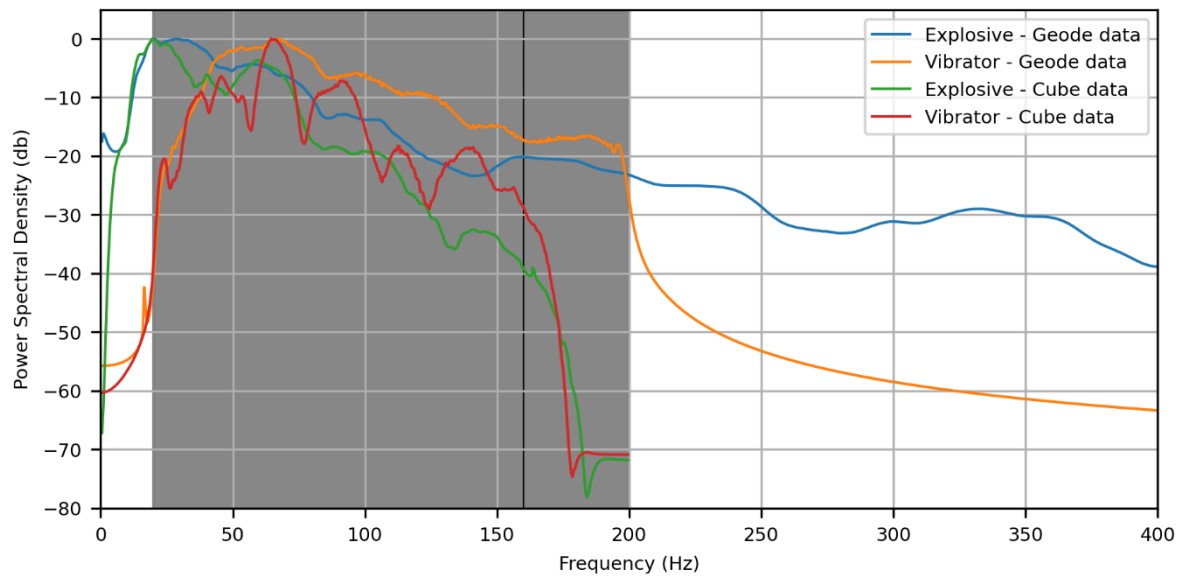


**Figure 5: Power spectral density for one record at profile SL-1P (corresponding record in Fig. 4). Explosive source and 20 Hz geophones (blue), vibratory source MHV4P and 20 Hz geophones (orange), explosive source and vertical component of 4.5 Hz geophones (green), vibratory source and vertical component of 4.5 Hz geophones (red). The sweep frequency range is shaded (gray).**

At the central part of SL-2P, we had to use ELVIS-7 as the vibrator source. The ELVIS-7 source (peak force of ~1 kN) emits less energy than the 4-ton hydraulic vibrator MHV4P (peak force of 30 kN) emits, which can directly be observed in the data (Fig. 6). While the first arrivals can be clearly traced along the entire receiver spread for MHV4P (blue arrows in Fig. 6), the first arrivals of the vibrator ELVIS-7 are only visible for offsets <250 m (orange arrows). The emitted energy of the ELVIS-7

source is less than the energy of the MHV4P, as we directly observe in the data. However, both sources show a frequency content for the entire sweep from 20-200 Hz above the background noise level (Fig. 6c).

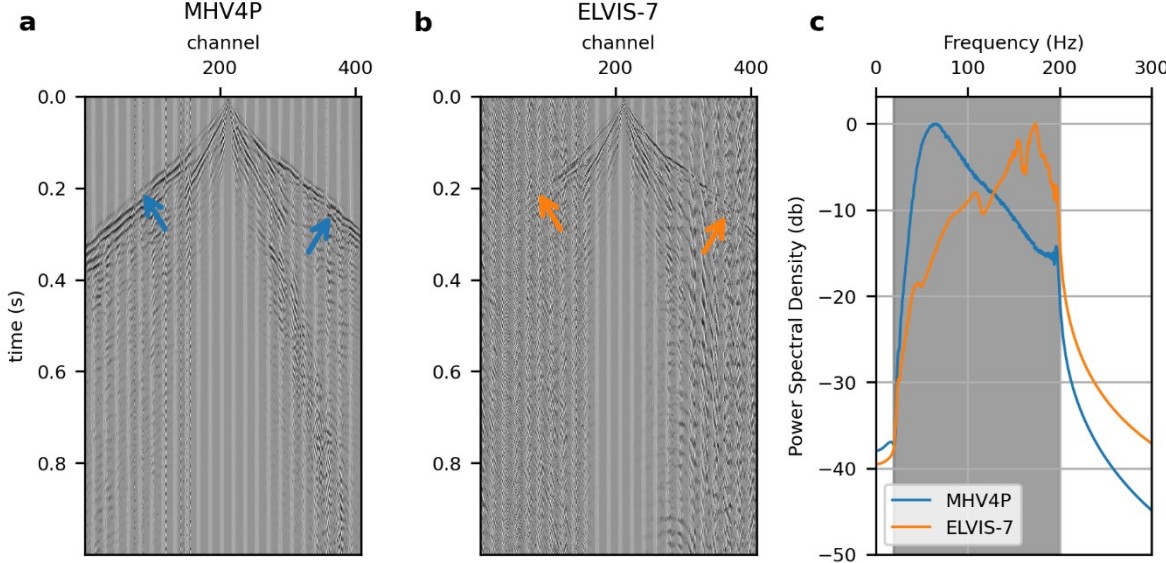

**Figure 6: Records of geode-data for neighbouring source locations for vibrators: (a) MHV4P and (b) ELVIS-7, and (c) corresponding power spectral density for vibrators MHV4P (blue) and ELVIS-7 (orange); the sweep frequency range is shaded (gray). The different source strength can be observed by first arrivals that are visible along the entire profile for MHV4P (blue arrows), but not for ELVIS-7 (orange arrows).**

Similar to the presented P-wave data, S-wave data show a good data quality (not shown). First arrivals and reflections can be observed on most records.

## 5.2 HRSR P-wave stacks

The newly acquired data utilizing the vibrator sources and geode-data image the basin base and internal reflectors (Fig. 7). The eastern profile SL-3P shows only horizontal reflectors, similar to the eastern end of SL-2P, and is therefore not shown. On all profiles, we observe horizontal reflectors below 300 m final datum that we interpret as Molasse units (red arrows). These reflectors can also be found at a shallower depth of the eastern part of SL-2P (pink arrows) and on the entire profile SL-3P (not shown). Thus, we infer that in the eastern part of the study site no basin is present, but that it is dominated by Molasse units. We interpret strong reflectors (green arrow) as basal till at the basin base. A dipping reflector within the basin (blue arrows) separates two generations of basin fill, which was previously unknown. Further internal reflectors are visible as well (purple arrows). The log of borehole 5068_3_A fits the reflectors and supports the interpretation. Unfortunately, the borehole can only support the interpretation for one part of the bipartitioned basin. In the part of SL-2P where we deployed the weaker ELVIS-7 source (between SL-1P and SL-3P in Figs. 7b, d), we observe shallow reflections (orange arrow), but cannot trace the deeper reflectors (yellow arrows) due to less penetration. Therefore, the eastern rim of the basin, that is the transition of the basal basin (green arrows) towards the shallow Molasse reflectors (pink arrows), is not imaged.

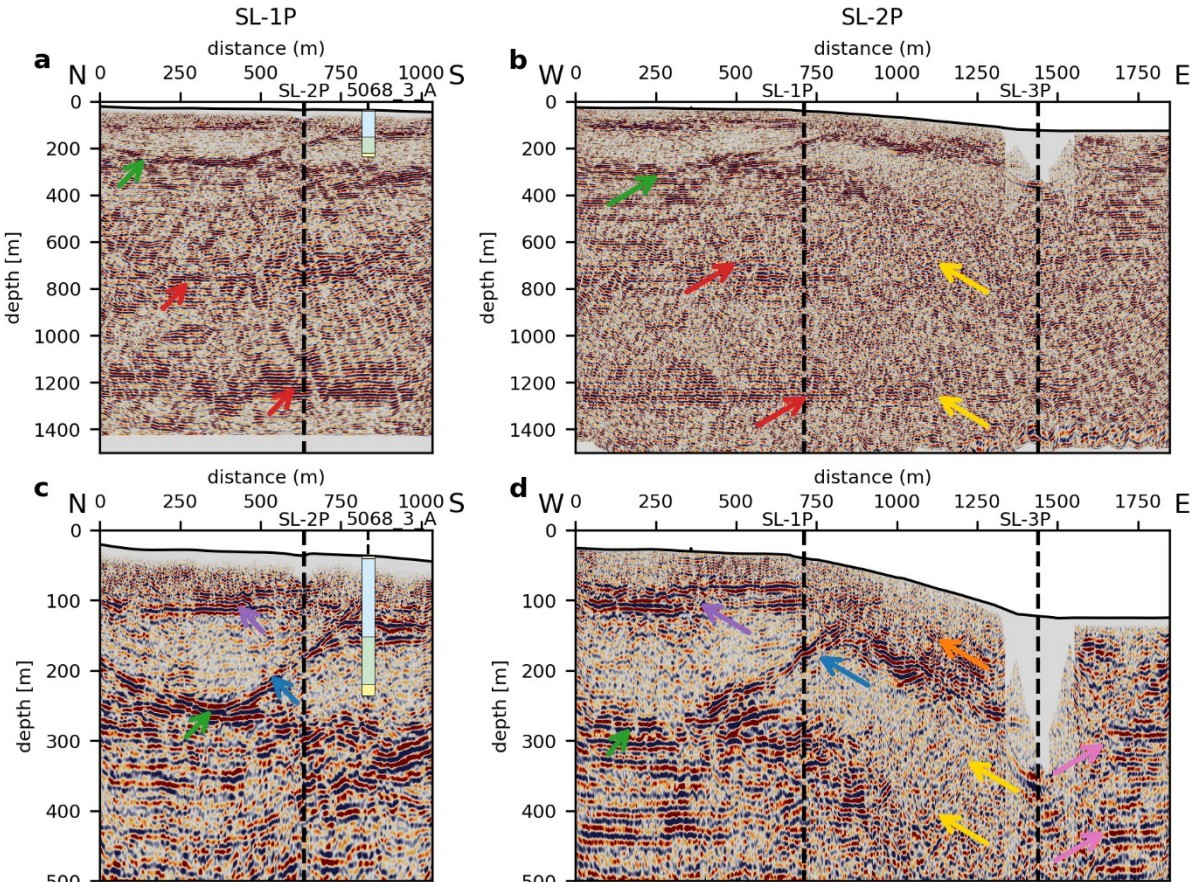

**Figure 7: Migrated stacks of (a) SL-1P and (b) SL-2P and zoom to the first 500 m of the same profiles with 1.6x vertical exaggeration (c, d). Highlighted by arrows are horizontal Molasse reflectors (red), shallow Molasse reflectors (pink), basin base (green), discontinuity of bipartitioning (blue), basin internal reflectors (purple), shallow reflectors (orange), and areas with less penetration (yellow). The ELVIS-7 source was deployed on SL-2P only, between SL-1P and SL-3P. The log of borehole 5068_3_A (cf. Fig. 1c) shows gravel (blue), fines (green), sand (orange), and diamict (gray).**

## 5.3 S-wave stacks and velocities

Stacks of the S-wave data show reflections from a few ms to about 700 ms two-way traveltime, generated by a small-scale ELVIS-7 source. The sections show a similar structural image of the subsurface but with some differences (Figs. 8a, b) compared to the P-wave images (Fig. 7):

1.   The reflections are less continuous compared to the P-wave data.
2.   S-wave data have less penetration depth than P-wave data.
3.   S-wave data show a much higher resolution than the P-wave data.

In detail, we interpret a strong reflection (green arrows) as basal till at the basin base. This reflection shows an undulation that cannot be seen in the P-wave data. A reflection separates the bipartioned basin

infill, as seen in the P-wave data (blue arrows). This also exhibits more details than the P-wave data, but is also less continuous. A basin-internal reflection is visible in the S-wave data (purple arrows) that is not present in the P-wave data. In the very shallow part, we observe a continuous reflection (red arrows) with a second reflection directly underneath (pink arrows). Both can be seen in the P-wave data as well, but S-wave data clearly show an overlap of this reflection (pink arrows) that is not visible in the P-wave data. The high resolution of the S-wave data enables detailed interactive velocity picking. S-wave interval velocities (Figs. 8 c, d), calculated from the stacking velocities, show lateral variations that match the overall geology.

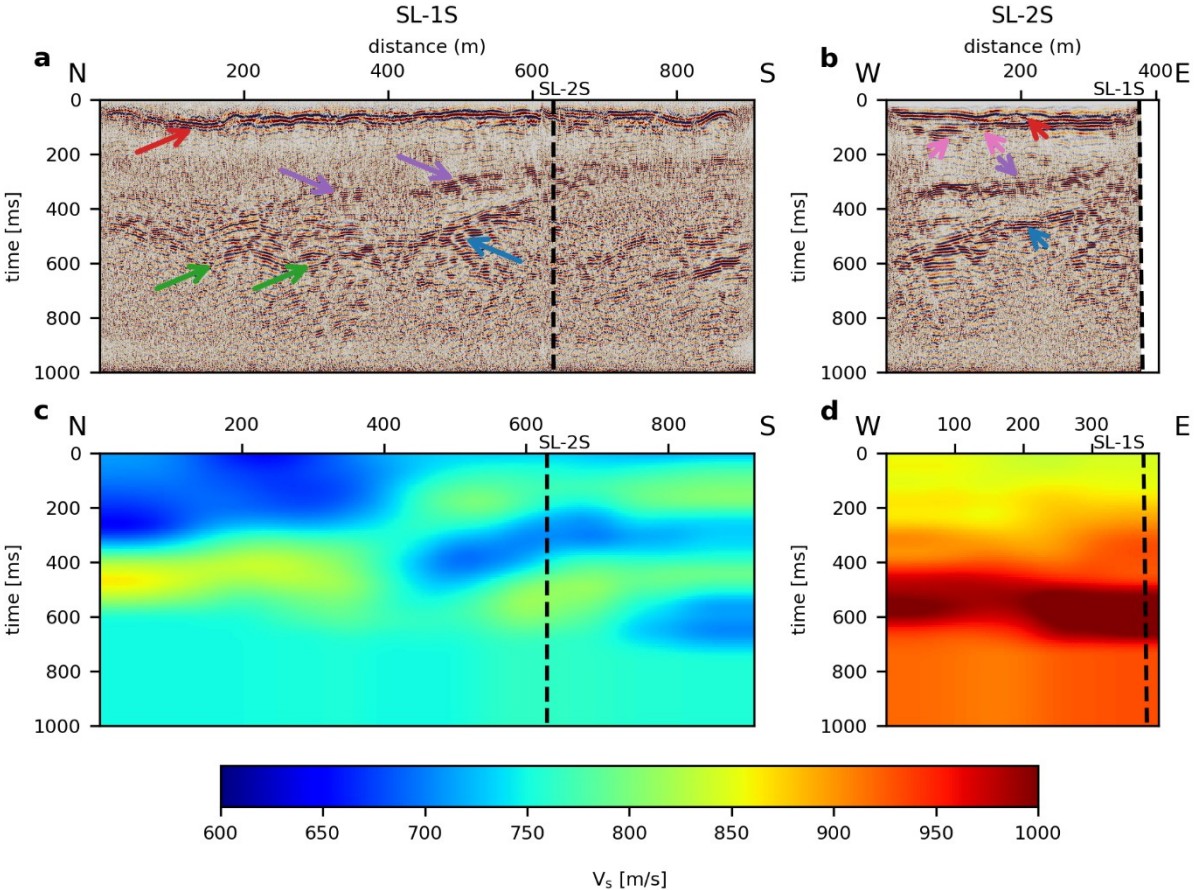

**Figure 8: Stacks of (a) SL-1S and (b) the western part of SL-2S and S-wave interval velocities of (c) SL-1S and (d) the western part of SL-2S. Highlighted by arrows are: basin base (green), basin-internal reflectors (purple), intra-basin discontinuity (blue), shallow reflections (red), and shallow discontinuity (pink).**

## 5.4 Full-waveform inversion

The different stages of inversion of the explosive source and geode-data show a successive updating of the velocity and density distributions from the initial 1-D gradient models to the most detailed model of the last stage, inverting frequencies from 7 Hz to 30 Hz (Fig. 9). Distributions of changes of the last

stage compared to the initial model reveal the potential of FWI. The P-wave velocity distribution (Fig. 10a) reveals an undulation in the velocities, which represents small-scale lateral variations in the velocity distribution (red arrows). The velocity results fit the reflectors of the stacked section (Fig. 10b; purple arrow). However, the FWI results contain details up to 30 Hz, while the stacked section contains data up to 200 Hz. The P-wave velocities of the FWI are in the similar range of the P-wave migration

velocity field, derived from HRSR processing (Fig. 10c).

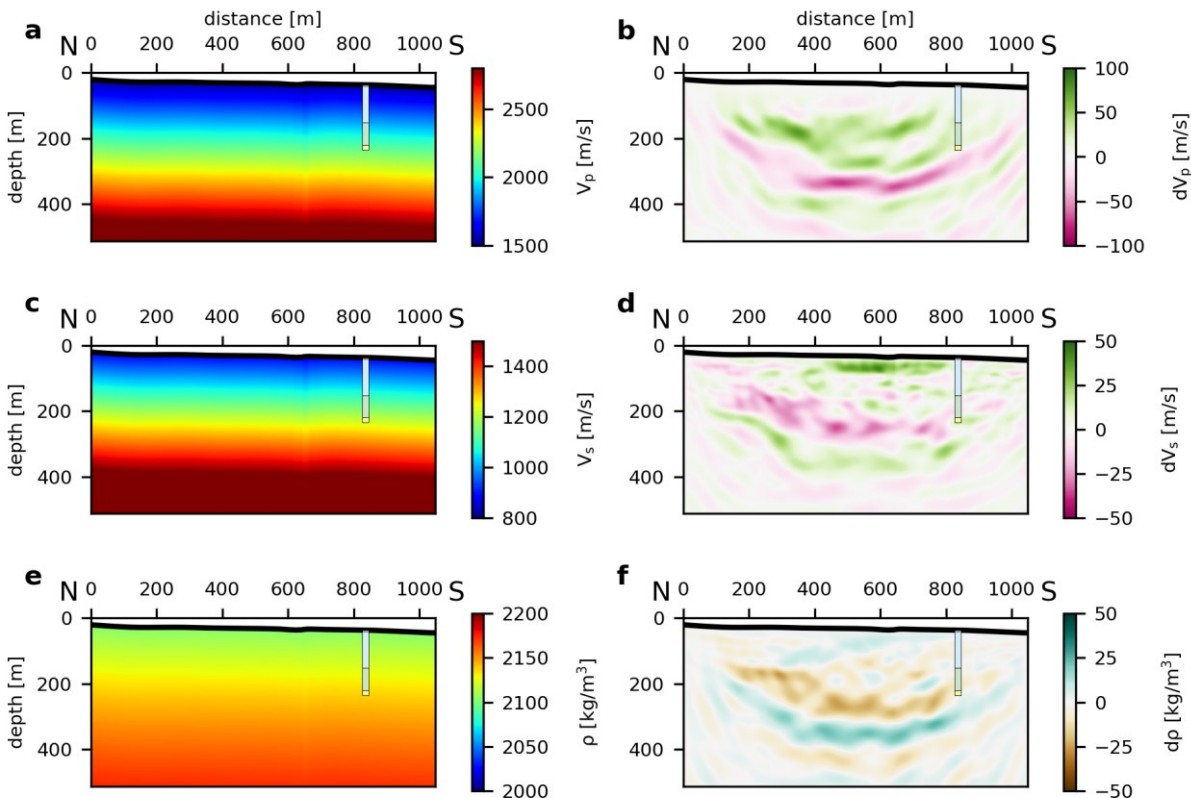

**Figure 9: Initial models and changes of the FWI of SL-1P to the initial model for (a, b) P-wave velocity, (c, d) S-wave velocity, and (e, f) density.**


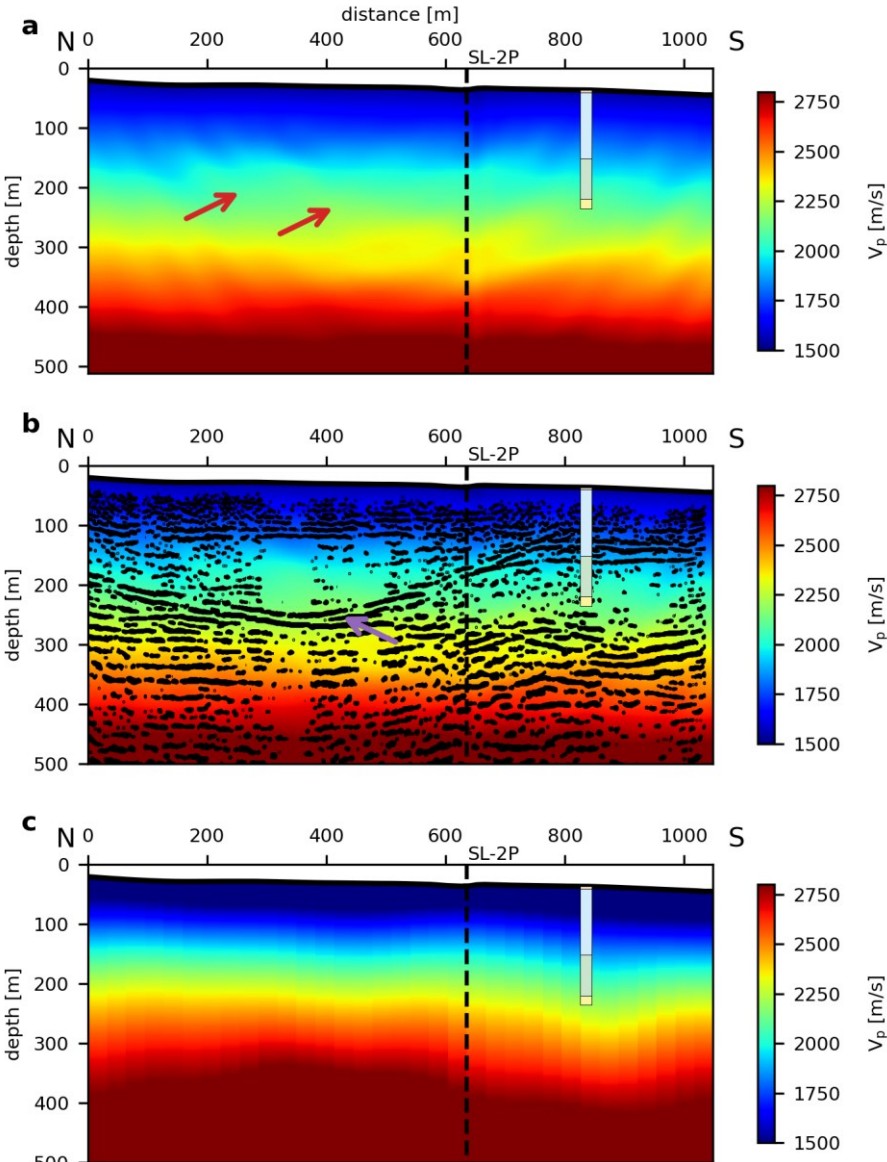

**Figure 10: Distributions from (a) the last stage of FWI of SL-1P for P-wave velocity, (b) FWI P-wave velocity with seismic section superimposed, and (c) P-wave velocity from HRSR processing. Highlighted by arrows are: undulations (red arrows) and matching with reflectors in the seismic P-wave section (purple arrow).**


## 6 Discussion

The dataset provides a valuable basis for developing the combination of HRSR and FWI for near-surface seismic imaging. The survey parameters were chosen to cover both requirements for HRSR as well as FWI analysis. For HRSR, we used parameters that worked well in Quaternary environments in

the past (e.g., Burschil et al., 2018; Buness et al., 2022). The lower sweep frequency of the vibrator data was determined by the technical limits of the available vibrator MHV4P. We chose the upper sweep frequency (200 Hz) due to damping and thus limited benefit of higher frequencies. The charge of the explosives was specified by the blaster. The previous refraction survey of the Bavarian Environment Agency in 2018 used charges of 250 g but did not penetrate through the unit of coarse-grained
sediments. This was the reason to increase the charge to 1 kg. The blaster split the 1 kg charge into four 250 g charges per borehole to avoid blowouts.

The lower frequency content of the cube-data is a valuable contribution to the further development of combining HRSR and FWI for the explosive sources. However, the geode-data (20 Hz geophones)
show a similar frequency content for the vibrator source as the cube-data (4.5 Hz geophones), so that the effort in this setup is questionable, if only vibrator sources are available.

A first attempt with explosive sources and the geode-data (20 Hz geophones) shows the potential of FWI to image sedimentary deposits in these environments. The forward-modelled data of the last stage
match the field data without cycle skipping (Fig. 11), so that the FWI works sufficiently for the acquired data. However, FWI is often applied to marine data, which often have a better signal-to-noise ratio. Sporadic other land studies also show the successful application of FWI (e.g., Köhn et al., 2012). On land, horizontally polarized S-wave data are often inverted to simplify the inversion problem (e.g., Schwardt et al., 2020; Köhn et al., 2019; Mecking et al., 2021). The penetration depth of the Rayleigh
wave is ca. one local S-wavelength (Sheriff and Geldart, 1995). For an average S-wave velocity of 1300 m/s, we get a maximum penetration depth of ~43 m at 30 Hz and ~185 m at 7 Hz. However, as can be seen in the waveform comparison of the records (Fig. 11), the waveforms of first arrival refraction and diving waves are also fitted, so the maximum resolution depth extends to ~ 0.5 * maximum offset of the acquisition geometry (~400 - 500 m). The applied staged FWI workflow of this study converges
successfully, provides a consistent and physically reasonable solution, and matches the HRSR results.

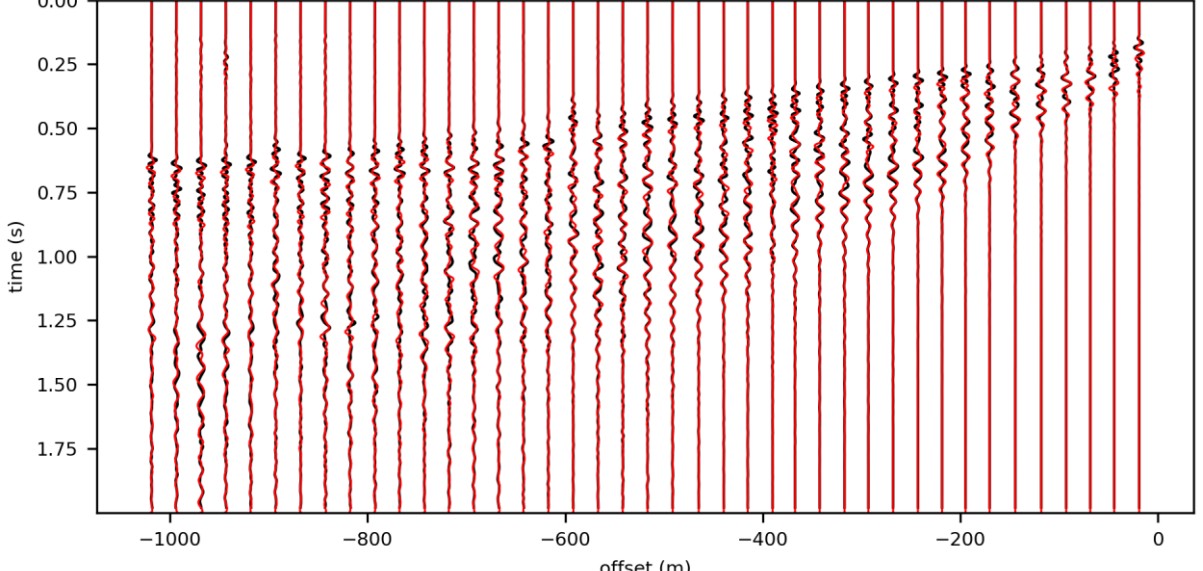

**Figure 11: Example of field data (black lines) and forward modelled data of the last stage (red lines) of one records. Modelled traces match the field data.**

A preliminary interpretation of the P-wave data shows detailed structure within the basin fill. Our interpretation fits the interpretation of the seismic sections, boreholes, and luminescence data by Firla et al. (submitted) at the same site. In general, the sedimentary sequence of glacial and post-glacial deposits is typical for these environments (e.g., Schaller et al., 2023; Schuster et al., 2024). The interpretation suits the reflection pattern that is similar to the seismic facies interpretation of the Tannwald Basin

(Burschil et al., 2018), consisting of different lithological units. The S-wave data show very detailed results, imaging the entire basin fill in very good quality, even using the small-scale source. The images show a much higher resolution than those derived from the P-wave data and reveal reflectors that were not observed in the P-wave data. This is in accordance with results from other studies (e.g., Pugin et al., 2009; Brodic et al., 2018; Burschil and Buness, 2020; Pertuz and Malehmir, 2023), even though P-

waves are commonly used to image Quaternary sediments for decades (e.g., Hunter et al., 1984; Büker et al., 1998; Maries et al., 2017).

## 7 Conclusions

The acquired datasets are suitable for the methodical development of combining HRSR and FWI. To meet the prerequisites of each method, the selection and integration of different source-receiver

configurations was essential. While vibrator sources with a broad frequency spectrum provide HRSR imaging, explosive sources generate low-frequency signals for FWI. The dense source and receiver spacing result in a valuable data acquisition scheme. The autonomous DATA-CUBE[3] record the low-frequency content of the explosive sources but do not give a benefit for vibrator sources in this study. Separate analyses of HRSR as well as FWI of different datasets show that each method provides

detailed images in the complex glacially influenced environment of the overdeepened basin.

HRSR images the basin structures as well as internal reflectors of the overdeepened basin close to the town Schäftlarn. We observe horizontal Molasse units and can delimit the eastern extent of the basin. The internal reflectors reveal a bipartioned basin that was previously unknown. S-wave data show more

details within the basin infill than P-wave data. This study provides valuable insights for future seismic investigations in glacially overdeepened basins and other heterogeneous geological settings.

## Acknowledgments

The project Chatseis is affiliated with the ICDP project Drilling Overdeepened Alpine Valleys (DOVE) and funded by the Deutsche Forschungsgemeinschaft (DFG, German Research Foundation) –

497340281. Particular thanks go to our team at BGR, LIAG, and LfU, especially the blasters Christian Veress and Brian Kröner, during the surveys, as well as the support of the municipality of Schäftlarn and the Monastery Schäftlarn.

## Code/Data availability

Acquired data are available under doi:10.25928/960y-8w55. DENISE Black edition is available at
https://github.com/daniel-koehn/DENISE-Black-Edition.

## Author contributions

TB managed the project Chatseis, including fieldwork organization, data acquisition, processing of S-waves, seismic interpretation, and preparation of the manuscript. DK organized the DATA-CUBE³ and performed FWI. MK processed the P-wave data. GG organized LIAG fieldwork, and JG organized LfU fieldwork and explosive sources. GF and MF conducted the geological interpretation. All authors contributed to the manuscript.

## Competing interests

The authors have no competing interests.

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
