# Peer review of "Seismic data acquisition for combining high-resolution seismic reflection and full-waveform inversion – a case study for overdeepened valleys"

_EGUsphere, 2025_

## Author Response (AR1)

**RC1**: 'Comment on egusphere-2025-2370', Samuel Zappalá, 14 Aug 2025 reply

RC1: Dear authors and editor,
the presented manuscript is an interesting case study. It analyzes an acquisition setup to acquire data suitable for both high resolution seismic reflection and full waveform inversion, highlighting the potential of simultaneous data acquisition. The topic well fits the scope of the issue.

AC: We thank the reviewer Samuel Zappalá for his review and assessment of the manuscript.

RC1: Introduction is clear and detailed. The description of acquisition and processing is confusing and will benefit of a more schematic organization of the presented information, especially considering the multiple source and recording systems employed. The discussion is poor and not fully clear, appearing mostly as a continuation of the results. Conclusions are quite general but reasonable for a case study.

AC: According to the following comments of RC1 and RC2, we reorganized the manuscript and reordered the sections. We moved the site description before the Methods section and combined the description of DOVE in this section.
Several tables, e.g., for the processing parameters, are included.

RC1: Following you can read my specific comments suggesting missing information and unclear parts on the text, and technical comments regarding some minor errors in the text.

Specific comments

The scope is a bit ambiguous, in the title and abstract you state that the scope is the imaging of the area with the method, while in the introduction you state "The scope of this paper is to demonstrate of suitable data acquisition and to create benchmark results for the combination of HRSR and FWI." where it looks like more focused on the method than on the results itself. It is perfectly fine to have both of them as scope if you like, but make it clear and subsequentially clearly structure the manuscript. As it is now, this ambiguity is present several times along the paper.

AC: We focused the scope of the manuscript. Accordingly, we changed the title and the statement in the Introduction.

RC1: Line 96 – previously you explain in detail different source and receiver types, but you never refer to passive seismic before this line, you could add a short sentence to introduce the method (i.e., employing background noise as energy sources).

AC: It was not our intention to refer to passive seismic techniques, therefore we clarified this phrase.

RC1: Line 102-103 – It is not clear what is the source point spacing used for the ELVIS-7 source and for the MHV4P vibrator. Is it the same for both? Rephrase this point or add the information if missing.

AC: Yes, it is the same source spacing for both sources. We add this information.

RC1: Line 104 and 110 – State the length of the sweeps and if they are linearly increasing.

AC: We added this information.

RC1: Line 114 – the receiver section (2.2) is very confusing. Organize chapter 2.2 better in a more schematic way, give the same information for all sensors, maybe add a table to make it visually clear. Add a section at the beginning of chapter 2 where you state all general acquisition information such as: how many profiles you are acquiring, how long are they, what your target depth is, etc., so that the reader can follow better what is happening.

AC: We move the site description prior to the data acquisition. In section data acquisition, we included a subsection about general information for the profiles and target depth as well as a table for additional profile information including profile length.

RC1: First you say you have 400 plus geophones every 2.5 m (>1000m), then you say that to keep 200 m of maximum offset in both direction (400 m spread) you need to use roll along geometry, then you say that you connect 24 geophones to Geodes recording units, are these different geophones or are you still referring to the same ones? Then you talk about three component geophones. Then you talk about DATA-CUBE, but to which one is it referred to? And then you have the landstreamer for S-waves recording, but you do not say which spacing you use, are they collocated with the vertical geophones? How are all these sensors located respectively. In addition to this you still did not show any map of the area or of the lines, no range of acquisition stated, and it is difficult to understand the scale of your project and your target.

AC: We rephrased this paragraph and smoothed the roll-along description. The 200 m refer to the minimal maximum offset. A table give additional specifications of the seismic receivers.
We reorganized the section by (1) recording unit, (2), receiver, (3) layout for P-wave and S-wave. A new paragraph separates the receivers.

RC1: Line 127, section 2.3 – give more details about the processing, which frequencies did you filter? Which AGC window did you apply? Which velocities range did you detect? Did you apply refraction static correction to S and P processing? You can add a table with the most important processing details if you prefer.

AC: We reorganized the processing into tables with additional parameters.

RC1: Line 136 – what is this profile you refer to? Still no indication of the acquired profiles.

AC: The same workflow and parameters were used for all P-wave profiles and a second workflow was used for all S-wave profiles. We add this information in the text.

RC1: Line 147, section 2.4 – this chapter is missing some detail (which data do you use in input for the FWI? The CUBE or the geode data?).

AC: Line 149-150 states "Input data were the explosive sources and the geode data." We extended this phrase to make it more prominent.

RC1: Line 151 – is a 1D starting model accurate enough for the FWI since it is usually very dependent on the initial model? You should show your starting model.

AC: In our case, the 1-D starting model is sufficient for the FWI. The staged workflow in combination with this initial model fits the data. The fit is shown in Fig. 11.
We added the initial models to the figure that shows the changes of FWI. For a better readability, we switched Figs. 9 and 10.

RC1: Line 158, Section 3 – move it before the actual section 2 and add the information about the acquisition that I said in my previous comment (how many profiles you are acquiring, how long are they, what your target depth is, etc.), then it will be easier for the reader to follow the acquisition setup information.

AC: We reorganized the manuscript and switched sections 2 and 3.

RC1: Line 189 – You could show only the first second of data so that the area with signal is more visible in the figure. You should show also a processed shot gather at some point in the manuscript before the stacked sections.

AC: We change the maximum time to 1 s. However, we did not show the processed gather to overload the manuscript.

RC1: Line 230, figure 7 – The colors shown in the borehole are quite difficult to relate to what you wrote in the caption, especially the "diamicton" that you state is gray but looks like yellow. Also, the magenta color in the arrows looks more like pink (maybe it is just the version for reviewers but check on it). It may be interesting to indicate on top

of the stacked section which source was used, since you never show it in the paper (a colored line with 2 different coded colors above the section should do the job).

AC: We checked the colors and emphasized the borehole log by adding an edge color. However, we stick to the colors of the detailed logs shown in Fig. 2, since these are the standard colors of the literature as used in Firla et al. (submitted).
For the P-wave survey, the ELVIS source was only on profile SL-2P in the central part between SL-1P and SL-3P. We added this information in the caption and the text.

RC1: Line 257 – what do you mean by "undulations in the velocities"? describe it in more detail the first time, then you can keep indicating it with the same term if you like.

AC: We added a clarification what we mean by undulations in the velocities.

RC1: Line 267 – I think you should also show your starting model since it is very important in the FWI.

AC: see previous comment: We included the initial models in new Fig. 9.

RC1: Line 275-285 – The information regarding why you choose certain acquisition parameters should be part of the acquisition setting and not of the discussion. You can discuss particularly innovative settings and their reasons in detail in the discussion, but you should not include traditional ones.

AC: Indeed, standard/traditional parameters that are commonly used should be part of the methods section. However, the reasoning for the chosen parameters, if mentioned, shall be in the discussion section. We move the standard parameters to the methods section but kept the reasoning in the discussions.

RC1: Line 287-291 – From "The S-wave data" to the end of the paragraph it is not clear your message and what you want to highlight or discuss, please rephrase it.

AC: We rephrased this paragraph.

RC1: At the end of the study, it is not clear what you use the cube data for. More in general, I would suggest writing with a more schematic approach all the different acquisition details for sources and receivers, why you choose them, for which method you use them (P, S, or FWI), and finally, using the same scheme, discuss whether these reasons are confirmed or not.

AC: We added a paragraph of the purpose of the cube data in the discussion. The cube data have not been used for HRSR or FWI so far. but are still a valuable dataset for the methodical development.
We have reorganized the Methods section to a more schematic approach as suggested.

RC1: I agree that the velocity model kind of agree with the seismic sections, but, before stating that it is an improvement towards the real velocity model, I would like to see: (1) the starting model; (2) the obtained velocity model from some other standard method; (3) some comparison with expected velocities, maybe from the borehole if available, or from literature in the area or for similar lithologies. At least something to show that the velocities are converging towards probably correct values and not towards a local minimum dependent on your starting model.

AC: (1) We included the initial models in new Fig. 9. (2) We updated new Fig. 10 and show the velocity distribution derived from HRSR. (3) The velocities correspond to those of the previous studies in these environments.

RC1: In the discussion you should discuss what the FWI obtained results give as extra information in your case. Why should people bother of acquiring extra data? Discuss it more. How much more accurate are the obtained velocities from the FWI respect to a more standard and computationally easier refraction tomography model? Discuss it more for your specific case (and not on why FWI is better than tomography in general).

AC: It is not our purpose to evaluate the benefit of FWI. Here, we present the data for the methodical development. We focused the scope of the manuscript including the change of the title so that the results of each method do not need to be compared among each other. This will be the topic of the following companion paper.

RC1: Why did you not use the velocities obtained by the FWI to migrate the stacked section?

AC: So far, our purpose is to present the data for the data and not to combine the methods. This will be integrated in a companion paper.

RC1: Technical comments
Line 40-44 – Provide some references.

AC: We included several references that supplement the statements.

RC1: Line 75 – "we emitted signal is close to the ideal" I think you mean "the emitted signal is close to the ideal".

AC: We corrected that.

RC1: Line 85 – Maybe you can add a reference to the early 20[th] century deployment of geophones.

AC: We added the reference Dragoset (2005).

RC1: Line 90 – You can write (DAS) just after Distributed acoustic sensing since it is commonly called by its acronymous.

AC: We added DAS.

RC1: Line 249 – "enables a details interactive velocity picking" maybe you mean "enables a detailed interactive velocity picking".

AC: We corrected the typo.

RC1: Line 257 – correct "Figs 9a" with "Fig 9a".

AC: We corrected this.

RC1: Line 261, figure 9 – the borehole plotted on (a) and (b) seems to miss the last lithological unit, the "diamicton". You should add in the caption that the borehole is superimposed.

AC: We enhanced the visibility of the borehole log in all figures.

RC1: Line 267 – Remove "change".

AC: We removed "change".

RC1: Line 267 – again the plotted boreholes are different between (a), (b) and (c).

AC: We enhanced the visibility of the borehole log in all figures.

RC1: Line 286 – In a scientific paper let's avoid subjective adjectives as "spectacular".

AC: We change this phrase into "… show a very detailed result, …".

RC1: Line 293 – In the phrase "Our interpretation fits to the interpretation (Firla et al., submitted) at the same site" maybe you mean "Our interpretation fits to the borehole interpretation (Firla et al., submitted) at the same site"?

AC: Actually, Firla et al. has a different version of the interpretation, since he included more data. We included this information in the text.

RC1: Line 330 – Fix "Data are available under acquisition are available under".

AC: We fixed this.

RC1: References:
Anselmetti et al. is indicated as 2021 in the text and as 2022 in the reference list.
Bohlen 2002 is not present in the text.
Dokter et al. 2017 is not present in the text.
Fichtner 2010 is not present in the text.

Pan et al. 2023 is not present in the text.

Penck et al. 1909 is not present in the text.

Preusser et al. is indicated as 2011 in the text and as 2010 in the reference list.

Sullivan 1998 is not present in the text.

AC: We checked the references carefully and removed all references not in the text and corrected some typos.

**Citation**: https://doi.org/10.5194/egusphere-2025-2370-RC1

**RC2**: 'Comment on egusphere-2025-2370', Anonymous Referee #2, 18 Aug 2025

RC2: The manuscript by Burschil et al. presents an interesting field acquisition designed to support both high-resolution seismic reflection (HRSR) and near-surface full-waveform inversion (FWI) over an overdeepened Alpine basin, using multiple seismic sources and receiver types. The acquisition is thoughtfully designed within the constraints of the site and available equipment.

AC: We sincerely thank Referee #2 for the constructive feedback and the assessment of the manuscript.

RC2: The dataset is valuable and the workflows are broadly appropriate, but the paper in its current form appears fragmented, and the FWI component is not sufficiently specified to support the stated aims. Substantial reorganisation, clearer articulation of objectives, and additional quality control and validation are required for the study to reach the level of a benchmark reference.

AC: We have thoroughly reorganized the manuscript in response to the suggestions of both reviewers. We have also refined the objectives, aims, and scope of the manuscript to enhance clarity and focus.

**General comments:**

RC2: The title suggests the acquisition is tailored to imaging overdeepened Alpine valleys, whereas the manuscript's stated objective is to demonstrate a suitable data-acquisition design and provide benchmark results for combining HRSR and FWI on such terrain. I recommend revising the title to reflect this emphasis, and ensuring the manuscript's stated goal is explicit in the Introduction and carried consistently through the paper.

AC: We have revised the title to more accurately reflect the refined scope and emphasis of the manuscript.

RC2: The manuscript's structure needs improvement. At present it is difficult to follow and reads as if multiple contributors' sections were never fully combined.

AC: We have significantly reordered the manuscript. The site description now precedes the methods section, and the methods section itself has been restructured.

RC2: 1. Reorder the study area / project description.
Consider moving Section 3 earlier in the paper and merging it with the DOVE project description (currently starting around line 60). That information does not need to necessarily be in the Introduction. Consolidating it will keep all "data" part together

and let you describe the study area as a whole. If the site primarily serves as an example for the acquisition approach (rather than being the main scientific focus), say so explicitly in this section.

AC:   We thank the reviewer for this suggestion and reordered the manuscript accordingly. We think that it got a better structure in its current form.

RC2: 2. Re-think the sectioning and flow between Sections 2 and 4.
The methods and results should follow a clean logic: design -> acquisition-> processing -> results (HRSR, then FWI) ->  integration/validation->conclusion/discussion. Consider changing the section and subsection accordingly.  Consider also folding the relevant part of Section 3 (from ~line 172) into. There is repetition across sections; streamline and improve readability by removing duplicated descriptions.

AC:   Due to the reordering, we come close to the suggestion. We included tables for acquisition und processing parameters that support this logic. The information of the acquired profiles (previously from ~172 is the beginning of the methods section.

**General questions:**

RC2: Please define what you mean by high resolution seismic in this study. Under which parameter changes would the same survey be considered standard-resolution rather than high-resolution?

AC:   In this study, 'high-resolution seismic' refers to a dense acquisition layout combined with high excited frequencies (>100 Hz). Both is essential to resolve small-scale structures in the near-surface (<200 m).

RC2: Please clarify the objective of applying FWI in this study, its added value beyond standard seismic reflection, and the role it is intended to play in the interpretation.

AC:   We have clarified the scope of the manuscript. This paper presents a dataset and its acquisition for the methodical development of combining HRSR and FWI. The detailed combination will be part of a companion paper. Here, we demonstrate that FWI works for the dataset, without a direct comparison of the methods at this stage.

RC2: Sections specific comments:
Introduction:
Integrate part starting on line 45 with the upper part, those should not be a standalone part.

AC:   We integrated the paragraph in the previous one.

RC2: Clarify and consider briefly extending the requirements (e.g. offsets, sampling, amplitude preservation, noise removal) of both methods so that the subsequent objectives and design choices follow naturally.

AC: We give the information of the requirements in the introduction.

RC2: Sections 2 and 4:
Add an acquisition summary table. Standardize source and receiver names (use one convention throughout) and reference consistently elsewhere.

AC: We added tables for acquisition, receivers, and processing.

RC2: Add a geometry figure near 2.1/2.2 subsections. Show profiles, receiver patches, source points, and borehole locations. If the study area is already described, you can merge this with Figure 3. As of now, it's hard to track which profiles and configurations were used.

AC: Following the reordering, the map of the profiles and borehole locations is now before the Methods section. We believe a detailed chart of receiver patches and source points is not beneficial, as SL-1P and SL-3P use single spread layouts, and SL-2P has only 2 rolls. S-wave surveys, rolling about 10 times per profile for only 60 m, are also not well suited for such a chart. We have rephrased the receiver subsection to clarify the layout geometry and refer to the Survey report (Burschil, 2024) for more details.

RC2: Reorder the sections. Move subsection 4.1 to follow the survey descriptions, then present the processing workflows. Briefly explain how each processing choice targets features visible in the data (e.g., ground roll, statics, noise removal, migration)

AC: We have reordered the sections, and processing is now described directly after data acquisition. A one-to-one explanation of each processing step's intention, tool, and parameter would overload the manuscript, so we refer to previous studies.

RC2: Line 122: what is the receiver spacing on the landstreamer?

AC: We added this information, i.e. 1 m.

RC2: Subsection 2.3:
Consider adding a figure visually representing the processing workflow.

AC: We added a table with the workflow information.

RC2: You apply true-amplitude recovery and surface-consistent deconvolution, but then two AGC passes. Was amplitude preservation an explicit goal? If so, was any amplitude-dependent analysis intended? Was AGC applied pre-stack or post-stack (and with what window lengths)?

AC:  Amplitude preservation was not a goal for processing. We have added information about the processing parameters, including AGC application (pre- or post-stack and window lengths), in the tables.

RC2: Subsection 2.4
The processing subsection describe the processing of "the vibratory source and geode data as well as the S-wave data", but for FWI dynamite data were used, how was those processed/prepared for the inversion?

AC:  For the FWI applications shown in the paper, we used low-frequency data up to 30 Hz, excited by an explosive source. The high-frequency vibratory data will be incorporated in following work and are part of the companion publication.

RC2: What is the rationale for using a 1-D starting model? Is the velocity field sufficiently simple along the line? Why not using a first-break traveltime tomography for a 2-D model? Add a starting model/gradient representation to Fig 13.

AC:  The simple 1D initial model was sufficient to achieve a reasonable waveform fit for the low-frequency explosive field data at 7 Hz. We plan to use a more sophisticated initial model based on first-arrival traveltime tomography in future publications, when incorporating the high-frequency vibroseis data.

RC2: How is the Vs starting model build?

AC:  The Vs initial model is estimated from the Vp model by using the simple empirical relation: vs = vp / sqrt(3).
We added this information in the manuscript.

RC2: And is the density starting model taken directly from the logs?

AC:  No, the density initial model is also estimated by an empirical relation
rho = 1000 * (0.1055 * log(vs) + 1.3871)
We added this information in the manuscript.

RC2: From what I understand the FWI algorithm that is used here inverts the Vp and Vs simultaneously? How is the density inverted?

AC:  Like the vp and vs model, the density is also updated using adjoint state gradients within the l-bfgs optimization method. We rephrase the corresponding sentence.

RC2: State the frequency schedule, time/frequency windowing and regularization. List key FWI controls and information about any cross-parameter validation.

AC:  The FWI workflow consists of a sequential frequency approach, inverting field data up to 7 Hz, 10 Hz, 15 Hz, 20 Hz, 25 Hz and 30 Hz, respectively. The source wavelet for

each shot gather is computed by a stabilized Wiener deconvolution. Smoothness constraints are imposed by an anisotropic, spatial 2D Gaussian filter, which length is adapted to the local P-/S-wavelength, applied to the adjoint state gradients, respectively. In x-direction the gradients are smoothed over 1x local wavelength, in y-direction over 0.5x the local wavelength. Parameter cross-talk is mitigated by using quasi-Newton l-bfgs optimization together with parameter scaling, where the density updates are systematically decreased by a factor 0.5x compared to the seismic-velocity model updates.

RC2: Try to provide also misfit-vs-iteration plots and a residual-gather example.

AC: Misfit-vs-iteration plots primarily confirm FWI optimization convergence but offer limited additional insight into result quality for this context. We prefer the direct waveform comparisons of synthetic and field data in the shot gathers (Fig. 11). Residual gathers can show large residuals even with a reasonable phase fit, which might be misleading for overall model quality.

RC2: Line 207: Point out the lower peak force that can be directly observed.

AC: We rephrase the sentence and add the information of the peak force in the text.

RC2: Line 220: Why you choose not to show entire SL-3P profile?

AC: We wrote in line 226: The eastern profile SL-3P looks similar to the eastern end of SL-2P and is therefore not shown. We moved this information at the beginning of the subsection.

RC2: Figure 7: The log representation and colours are not visible on the figure, adjust the log and colour scale.

AC: We checked the colors and emphasized the borehole log by adding an edge color. However, we stick to the colors of the detailed logs shown in Fig. 2, since these are the standard colors of the literature as used in Firla et al. (submitted).

RC2: Line 244 and Figure 7 and 8: Consider adding a composite/side-by-side figure that aligns P and S stacks over overlapping segments. Annotate shared features and those visible only on S-wave data. Where feasible, add a third panel with FWI sampled along the same segment to tie reflectivity to elastic properties.

AC: Detailed interpretation and comparison of HRSR stacks are not within the current scope of this manuscript. Therefore, we decided against an additional composite figure showing P- and S-wave stacked sections to avoid over-complicating the paper. However, the revised Fig. 10 now compares P-wave velocities derived from FWI and HRSR, addressing the spirit of integration.

RC2: Subsection 4.4.
Please state explicitly and briefly justify to what depth you consider the Vp, Vs, and density models reliable.

AC: Penetration depth of the Rayleigh wave is ~ one local S-wavelength. So, for an average S-wave velocity of 1300 m/s, we get a maximum penetration depth of ~43 m at 30 Hz and ~185 m at 7 Hz. However, as can be seen in the waveform comparison of the shot gathers, we also fitted the waveforms of first arrival refraction and diving waves, so the maximum resolution depth is extending to ~ 0.5 * maximum offset of the acquisition geometry ~ 400 - 500 m.

RC2: Consider supporting the results with checkerboard resolution test.

AC: More extensive resolution tests will be a focus of upcoming papers, especially when inverting high-frequency vibroseis data. A detailed resolution test or discussion of the velocity distribution is beyond the scope of this manuscript.

RC2: At present, the Vp/Vs models appear only weakly correlated with structural features in the seismic sections, whereas the valley geometry is reproduced most clearly in the density model. Could you comment on possible causes?

AC: No, we will analyze this behaviour in more detail in the upcoming FWI of high-frequency vibroseis data with the FWI code and present it in the planned companion paper.

RC2: Figure 9: If feasible (are logs available?), add a borehole–FWI tie: overlay Vp, Vs, and density logs on 1-D extracts from the FWI model

AC: Unfortunately, there are no geophysical logs available. A detailed description of the borehole data will be published in Firla et al. (submitted).

RC2: Discussion/conclusions:
Conclude with a few explicit acquisition design takeaways others can reuse in Alpine basins.

AC: We added the takeaways in the Conclusions.

RC2: To proof the feasibility of integrating both HRSR and FWI try to demonstrate one concrete step (e.g., depth conversion of the P-stack using FWI Vp; or statics refined from near-surface FWI) rather than only discussing integration conceptually.

AC: In this paper, we focus on presenting the data and demonstrating its sufficiency for each method (HRSR and FWI) separately. The combination will be thoroughly addressed in a forthcoming companion paper.

RC2: Minor/technical comments:
Consider using only one of: high-resolution seismic reflection or high-resolution seismic method

AC: We avoid two phrases and use only high-resolution seismic reflection.

RC2: Line 75: "we" should be replaced with "the"

AC: We corrected the sentence.

RC2: Figure 6c: Correct "Frequenz" in the figure title

AC: We corrected the labels of Figs. 5 and 6c.

RC2: Correct "anti alias" to anti-alias or anti-aliasing

AC: We check the manuscript and use the term "anti-alias".

RC2: Line 198: Anti-alising filter is 80% of the Nyquist frequency, not sampling rate. Please correct.

AC: We corrected the sentence.

RC2: Figure 10: Correct the figure caption.

AC: We corrected the figure caption.

RC2: Ensure the borehole is clearly visible/legible on all figures.

AC: We enhanced the visibility of the borehole log on all figures.

**Citation**: https://doi.org/10.5194/egusphere-2025-2370-RC2

**EC1**: 'Comment on egusphere-2025-2370', Christopher Juhlin, 23 Aug 2025

EC1: Dear Authors,
The two reviewers are in agreement that the paper needs significant improvement in the clarity of the presentation. Please pay special attention to this when responding to their comments on a point by point basis.
Best Regards,
Chris Juhlin (Guest editor)

AC: Dear Editor, we have meticulously revised the manuscript, paying special attention to improving the clarity of presentation as suggested. We have provided a detailed, point-by-point response to all comments from both reviewers.

Kind regards, Thomas Burschil on behalf of all co-authors

**Citation**: https://doi.org/10.5194/egusphere-2025-2370-EC1

---

## Author Response (AR2)

**Report #1**, Submitted on 14 Oct 2025, Referee #1: Samuel Zappalá

RC1: Dear authors and editor,
The manuscript has been largely reorganized according to reviewers' suggestions and now it reads much better and it is clearer and easier to follow for the reader. The scope of the manuscript has been better defined, updating the title and better clarifying in the introduction the focus points of the study. The included tables effectively schematize and summarize different aspects of the acquisition setup. With the new setting the focus of the manuscript is clearly on the acquisition setup and on how this setup can favorite both reflection seismology standard processing and FWI analysis from a single acquisition (even if considering separate sources). The details of processing are not deepened since not considered as scope of the study as well as the FWI analysis that will be focus of a specific following paper.

AC: We thank Samuel Zappalá for the second review of the manuscript and agree that the scope of the manuscript is focused in the revised version.

RC1: Specific comments
Line 202-203 – The paper Burschil et al. (2018) shows a quite different processing while Denhert et al. (2012) almost do not show any processing. Still, if you don't want to put more details it is fine, the processing steps are quite standard and do not need extra explanation. The only step I am curious is the S10, in the table you write 200% stretch mute for NMO correction, I assume it is because of the low velocity of S-waves but maybe you can write a phrase about it.

AC: To avoid confusion, we deleted the references.
It is true that the 200% stretch mute for NMO correction is due to the low S-wave velocities. We added a sentence.

RC1: Figure 7 – I am sorry, I know you said in your response that you checked the colors for the borehole, but I still cannot relate to them. The added edge better defines them and now it is easier to distinguish them and relate it with the general borehole added in figure 1c. I am perfectly fine with you keeping these colors but the colors that you use in the figure and that you refer to in the caption should match, in the caption you say that there is blue, green, orange and gray but in the figure I see orange, blue, green and yellow. And also write them from the shallower to the deeper so it is easier to follow.

AC: We adapted the generalized dominant lithology of Fig. 1c in the caption of Fig. 7.

RC1: Figure 8 – Just an observation, the S wave velocities in profile 2 are quite faster than the ones in profile 1. Can you explain that? Do you think it is related to any anisotropy in the area?

AC: We cannot explain the higher velocities with anisotropy in the area. It needs further analysis of the data.

RC1: Line 344 – The stacked section should contain data up to 170 Hz since you bandpass filtered it.

AC: We corrected the upper frequency of the stacked section.

RC1: Figure 9 – In the density plot (9e), why are you using a colorbar starting from 2000 kg/m3 if your first values seem to be from 2100? If there is not a particular reason I suggest to fix it for a better comparison with the velocity models.

AC: We changed the color scale of Fig. 9e.

RC1: Line 408 – Do you mean "a benefit with respect to"?

AC: We corrected the sentence.

RC1: Technical comments
Figure 1 – Nice figure, but in the current size the text is too small, especially the labels in the borehole and the legend are not readable.

AC: We enlarged the Fig. 1.

RC:1 Table 1 – In the caption maybe you can state "vibrator (VP) and explosive (SP) shot points" since vibrator points are also a type of shot points.

AC: We adapted the phrase.

RC1: Table 1 – You may consider also to add the shot (5 and 4 m) and receiver (2.5 m) spacing in the table.

AC: We added the VP and RP spacings in the table.

RC1: Figure 4 – Showing one second of data works better and more details are now visible, but you forgot to move the arrows in figure b and maybe c.

AC: We corrected the positions of the arrows.

**Report #2**, Submitted on 27 Oct 2025, Anonymous referee #2

RC2: I thank the Authors for addressing most of my previous comments. The manuscript's structure and content have improved considerably. However, one more careful revision would further enhance the clarity, flow, and overall readability of the text.

AC: We thank reviewer 2 for the suggestions.

RC2: Please review the manuscript again to refine language, grammar, and transitions, especially in the Introduction (around line 48), where the flow remains uneven and some sentences need clarification.

Throughout the text—particularly in the Acquisition and Processing sections—many sentences are short and read as fragmented lists. Combine and smooth them for better narrative flow, and remove redundant parts.

AC: We edited the manuscript comprehensively and made the text smoother. A native speaker also edited the manuscript.

RC2: The Discussion section does not currently function as a true discussion but instead summarizes and justifies previously presented results, repeating content from the Conclusions. Please consider reflecting instead on what the findings imply for future combined HRSR–FWI surveys and how they can inform acquisition and processing design in future studies.

Conclude with concise, practical takeaways relevant to acquisition design for future studies, including those combining HRSR and FWI.

AC: We discussed our results with respect to other studies and give reasoning for the chosen approach. We revised the conclusion and give a take-home message.

RC2: Avoid unnecessary technical details (e.g., software version numbers) and maintain consistent past tense throughout, as both acquisition and processing have already been completed.

AC: We removed the version numbers of the software and revised the acquisition and processing section to past tense.

RC2: Minor comments:
   • Lines 54–56: Repetition regarding the importance of the initial model.

AC: We rephrased the sentence.

RC2: • Line 51: Should refer to S-wave velocities from vertical component data, not P-wave data.

AC: Were, we emphasize the importance of both types of initial velocity models for FWI, not only the S-wave initial model.

RC2: • Line 48: "Fertilizes" is an awkward word choice; replace with improves, refines, or enhances.

AC: We changed the wording.

RC2: • Line 238: Replace "whose".

AC: We rephrased the sentence.

RC2: Table 3: Consider integrating the table information into the text or clarifying its purpose.

AC: We added this table to give the acquired data more schematic organization, as supposed by the reviewer comments.

**Associate editor decision: Publish subject to minor revisions (review by editor)** (by Christopher Juhlin):

CJ:  Public justification (visible to the public if the article is accepted and published): The authors present a nice case study using an acquisition procedure that allows for both reflection seismic processing and full waveform inversion to be applied to the data acquired over an overdeepened valley.

AC:  We thank Chris for handling the review process and included all comments of the reviewers.

**Notification to the authors (**by Mario Ebel):

ME:  1) Please ensure that the colour schemes used in your maps and charts allow readers with colour vision deficiencies to correctly interpret your findings. Please check your figures using the Coblis – Color Blindness Simulator (https://www.color-blindness.com/coblis-color-blindness-simulator/) and revise the colour schemes accordingly. --> Fig. 5

AC:  We adapted Fig. 5.

ME:  2) Your reference list includes works "submitted". Such works can be cited upon submission if being available to the reviewers. They cannot be cited in the final, accepted manuscript, unless published, accepted for publication, or available as preprint with a DOI.

AC:  Since the manuscript is not published yet, we changed Firla et al. (submitted) to Firla et al. (2024).